# Selective Genotyping and Phenotyping for Optimization of Genomic Prediction Models for Populations with Different Diversity

**DOI:** 10.3390/plants13070975

**Published:** 2024-03-28

**Authors:** Marina Ćeran, Vuk Đorđević, Jegor Miladinović, Marjana Vasiljević, Vojin Đukić, Predrag Ranđelović, Simona Jaćimović

**Affiliations:** 1Laboratory for Biotechnology, Institute of Field and Vegetable Crops, National Institute of the Republic of Serbia, Maksima Gorkog 30, 21000 Novi Sad, Serbia; 2Legumes Department, Institute of Field and Vegetable Crops, National Institute of the Republic of Serbia, Maksima Gorkog 30, 21000 Novi Sad, Serbia; vuk.djordjevic@ifvcns.ns.ac.rs (V.Đ.); jegor.miladinovic@ifvcns.ns.ac.rs (J.M.); marjana.vasiljevic@ifvcns.ns.ac.rs (M.V.); vojin.djukic@ifvcns.ns.ac.rs (V.Đ.); predrag.randjelovic@ifvcns.ns.ac.rs (P.R.); simona.jacimovic@ifvcns.ns.ac.rs (S.J.)

**Keywords:** soybean yield, genomic selection, single nucleotide polymorphism, training set, soybean breeding

## Abstract

To overcome the different challenges to food security caused by a growing population and climate change, soybean (*Glycine max* (L.) Merr.) breeders are creating novel cultivars that have the potential to improve productivity while maintaining environmental sustainability. Genomic selection (GS) is an advanced approach that may accelerate the rate of genetic gain in breeding using genome-wide molecular markers. The accuracy of genomic selection can be affected by trait architecture and heritability, marker density, linkage disequilibrium, statistical models, and training set. The selection of a minimal and optimal marker set with high prediction accuracy can lower genotyping costs, computational time, and multicollinearity. Selective phenotyping could reduce the number of genotypes tested in the field while preserving the genetic diversity of the initial population. This study aimed to evaluate different methods of selective genotyping and phenotyping on the accuracy of genomic prediction for soybean yield. The evaluation was performed on three populations: recombinant inbred lines, multifamily diverse lines, and germplasm collection. Strategies adopted for marker selection were as follows: SNP (single nucleotide polymorphism) pruning, estimation of marker effects, randomly selected markers, and genome-wide association study. Reduction of the number of genotypes was performed by selecting a core set from the initial population based on marker data, yet maintaining the original population’s genetic diversity. Prediction ability using all markers and genotypes was different among examined populations. The subsets obtained by the model-based strategy can be considered the most suitable for marker selection for all populations. The selective phenotyping based on makers in all cases had higher values of prediction ability compared to minimal values of prediction ability of multiple cycles of random selection, with the highest values of prediction obtained using AN approach and 75% population size. The obtained results indicate that selective genotyping and phenotyping hold great potential and can be integrated as tools for improving or retaining selection accuracy by reducing genotyping or phenotyping costs for genomic selection.

## 1. Introduction

Soybean (*Glycine max* (L.) Merr.) is the most important protein crop used for animal feed and human food, industrial and bioenergy purposes, containing numerous compounds with a positive effect on health, such as phytoestrogens and antioxidants. At current genetic gains, global soybean production will increase globally by about 55% by 2050, which is insufficient to meet the projected demands of an increasing global population [1]. Moreover, global soybean yields are predicted to decrease by 3.1% with each °C shift caused by climate change [2]. Thus, one of the main goals of soybean breeding programs is to accelerate the rate of genetic gain for yield. In this context, the application of advanced breeding techniques, such as genomic selection (GS), may be beneficial.

Genomic selection (GS) assumes the estimation of individual effects of genome-wide markers on quantitative traits using a training set (TS) [3]. The additive sum of estimated marker effects is used for the calculation of the genomic-estimated breeding values (GEBV) of a selection candidate. In the past ten years, numerous studies on soybeans have examined the potential of GS to enhance the breeding for biotic stress resistance [4,5,6], quality-related traits including amino acid concentration [7], seed sucrose content [8], and protein and oil content [9,10,11,12]. In addition, multiple studies investigated the possibility of using genomic selection in soybean breeding to increase the genetic potential for yield and yield-related traits, such as maturity, plant height, and 100-seed weight [10,11,12,13,14,15,16,17].

The improvement of prediction accuracy is an important issue for the practical application of genomic selection. Several factors affect the efficiency of prediction, including trait architecture and heritability, training set size and composition, genomic relationship between the training set and test set, marker density, linkage disequilibrium (LD) between markers, and statistical model for estimation of marker effects [18,19]. Genomic selection demands extensive genotyping for both the training set and selection candidates, which raises the overall program cost. However, several studies have reported that higher genotyping density does not always improve accuracy and decreasing marker density can have minimal impacts on prediction or sometimes outperform the entire dataset [14,20,21,22,23]. Thus, a minimal and optimal number of markers with high accuracy for the phenotype should be determined, which is, in practice, a challenging task [24]. Assessment of the optimal marker set reduces the dimensionality of genomic data for prediction and provides a small number of parameters for better generalization in prediction modeling [25]. Different approaches for obtaining optimal marker sets were evaluated, including random selection of markers, selection based on the distribution of markers across the genome, selection based on marker effect, selection based on haplotype block analysis, and genome-wide association studies [26,27,28]. Eliminating markers with a small effect could be a good strategy for establishing trait-specific low-density SNP chips which would consequently reduce genotyping costs in breeding programs [29].

The design and optimization of the training set is a critical step for the enhancement of efficiency and effectiveness of genomic selection as predictions are based on genotyping and phenotyping data of selected genotypes. Moreover, the most important constraint in plant breeding programs is phenotype information due to the high cost of phenotyping. Thus, the reduction of phenotyping cost and increment of the quality of the phenotypic data could be obtained by designing smaller TS [30]. In general, the improved prediction ability of a larger training set is a reflection of an increased replication of rare alleles which improves estimations of these marker effects [13], reducing their bias and variance [19]. However, using relatively a smaller TS was reported to be as accurate as using a larger training population [31]. Traditionally, selective phenotyping is mostly based on random sampling of a whole set of genotypes [13], although random sampling does not always increase predictive ability due to the under- or over-representation of the genetic information in the TS [19]. Selective phenotyping based on marker data might reduce the number of genotypes that need to be tested in the field while preserving the genetic diversity of the initial population. Populations with different structures contain different levels of linkage disequilibrium, so the ideal TS size for genomic selection may be population-dependent [11]. Furthermore, it is expected that fewer markers for genomic prediction will be needed in populations with a less pronounced structure, in less complex and less divergent populations.

High costs associated with genotyping and phenotyping of a large number of genotypes are some of the constraints that disable the wider application of genomic prediction in selective breeding programs. Thus, to speed up the application of GS, it is crucial to create cost-effective strategies by decreasing genotyping and phenotyping costs while increasing or retaining the precision of genomic prediction. This study aimed to evaluate different methods for the selective genotyping of the optimal marker set and selective phenotyping for the optimization of the training set, and their effects on the accuracy of genomic prediction models for soybean yield, trained for populations with different diversity and population structure.

## 2. Results

### 2.1. Predictive Ability across Different Methods for Selective Genotyping

Three soybean populations were used to evaluate selective genotyping and phenotyping strategies. The efficiency was expressed through the prediction ability, assessed in cross-validation (CV) and external validation (EV). Besides origin, the examined populations differed in the size of training and validation sets and the initial number of markers used (Table 1). In all populations, the initial marker sets were first pruned to minimize the influence of SNP clusters in linkage disequilibrium. In this way, the initial marker number was reduced to 1044, 20,242, and 22,470 SNPs in RIL (recombinant inbred lines), MDL (multifamily diverse lines), and GPL (germplasm collection lines) populations, respectively (Table 1). Furthermore, by applying different selective genotyping strategies, five additional subsets of markers were derived and evaluated (6072, 1821, 546, 163, and 48 SNPs) in MDL and GPL populations, and three in the RIL population (546, 163 and 48 SNPs).

In 10-fold cross-validation, the average prediction ability using all markers was 0.29 in RIL, 0.59 in MDL, and 0.72 in the GPL population. Generally, in cross-validation, all populations followed a similar pattern of prediction ability for different marker reduction strategies. Namely, strategies RE-MaB and NRE-MB showed a sudden increase of *r_cv_* values when using marker subsets for model development. Strategies RE-MoB and NRE-R in general had similar patterns, particularly in MDL and GPL populations, whereas in the RIL population, RE-MoB had higher values for prediction ability compared to NRE-R. This trend was constant for all sets of markers. When comparing these two approaches, it is important to emphasize that Figure 1 represents average values of 1000 repetitions for NRE-R and that minimal values are lower. In the RIL population, the RE-MoB approach selected 48 SNPs that could retain *r_cv_* values obtained using all markers. Furthermore, in the MDL population, predictive ability using 48 SNPs was 0.49 and was increased by 0.06 from 48 SNPs to 163 SNPs (Figure 1b). Utilizing 546 and more SNPs derived with RE-MoB did not significantly increase the *r_cv_* for yield. It maintained similar values for all subsets greater than 546 SNPs. For GPL, the changes in *r_cv_* followed a similar trend, with an increase from 0.59 (48 SNPs) to 0.65 (163 SNPs) and from 0.68 (546 markers) to 0.70 (1821 markers) (Figure 1c), and no significant differences were observed using larger sets of markers. To further examine possible strategies for reducing the number of markers for prediction, external validation was performed for the three populations (Figure 2).

In the external validation for yield in soybean, *r_ev_* values using all markers were 0.36 in RIL, 0.60 in MDL, and 0.41 in the GPL population. In the RIL population, a similar pattern of different strategies as in cross-validation was observed, with the highest values of prediction ability for RE-MaB and NRE-MB approaches. Furthermore, the RE-MoB strategy retained similar values of prediction ability for all subsets of markers, similar to cross-validation. In the MDL population, the highest values were obtained using the NRE-R procedure, followed by RE-MoB. Similar to cross-validation, NRE-R represented average values of 100 models with randomly selected markers. Minimal values of NRE-R for subsets of 6072 SNPs and lower went below values of prediction ability for the RE-MoB approach. Using this strategy, obtained marker subsets could retain high values of prediction ability up to a set of 163 SNPs. Only the subset of 48 SNPs showed a decrease in *r_ev_* (0.44). In external validation, strategies RE-MaB and NRE-MB showed the lowest performance, in both the MDL and GPL populations, opposite to cross-validation. This could indicate potential overfitting of given models in cross-validation and these strategies cannot be used reliably for marker selection. In the external validation with the GPL population, RE-MoB mostly had the highest values of prediction ability, followed by NRE-R. In all cases, RE-MoB had higher values compared to minimal NRE-R values. Thus, the subsets obtained by applying the RE-MoB strategy can be considered the most suitable procedure for marker selection for all populations.

Apart from the described strategies, GWAS-based marker selection was performed for genomic prediction in the MDL and GPL populations. This method was based on GWAS results, genotypic and phenotypic data, and using ridge regression best linear unbiased prediction (RRB) and bootstrap trees (BTS) selection methods. The FarmCPU method was used for the association study by iteratively using fixed and random effect models with the most significant markers incorporated as covariates. In this way, model overfitting is avoided, statistical power is improved, computational efficiency is increased, and false positives and false negatives are controlled [32]. Considering the differences in sample sizes, 3-fold and 5-fold cross-validation was used for the MDL and GPL populations, respectively. Thus, the training and validation sizes in cross-validations were 151 and 76 in MDL and 886 and 221 genotypes in the GPL population (Table 2). RRB and BTS selection methods selected 306 out of 34,889 and 1112 out of 38,184 markers for the MDL and GDL populations, respectively (Table 2). In both populations, selected markers were distributed throughout all 20 chromosomes. The average value of the correlation between observed and predicted phenotypes of the validation sets was 0.941 (MDL) and 0.886 (GPL) (Figure 3). BTS and RRB methods read all markers for all CVs in analyzed populations.

Further modeling for genomic prediction was performed using a selected marker set and four prediction models (RRB, random forest (RF), deep neural network (DNN), and convolution neural network (CNN)). After the establishment of models, the test sets of MDL and GPL populations, which were not included previously in the marker selection, were used for phenotype prediction, and the correlation between predicted and observed phenotypes was calculated. The model with the highest correlation rate on the validation set is considered the best final prediction model. In the MDL population, the RRB prediction model had the highest correlation rate of 0.98 in a validation set and thus was selected as the model with the best performance. In the test set of the same population, the correlation using the chosen model was 0.4, although the highest value for the test set was observed in the RF model (0.62). The obtained value was lower compared to the *r_ev_* values of the RE-MoB strategy and the smallest set of 48 SNPs. In the GPL population, the RRB prediction model also showed the best prediction performance (0.93) in the validation set, while in the test set, the performance of the selected model was 0.33. This value achieved with 1112 SNPs was comparable to the observed prediction ability in models using RE-MoB sets of markers where *r_ev_* ranged from 0.292 (546 SNPs) to 0.398 (1821 SNPs).

### 2.2. Prediction Ability across Different Methods for Selective Phenotyping

Prediction ability in both cross-validation and external validation for a random sample selection for the MDL and GPL populations plateaued around a TS size of 50% (Figure 4). Prediction ability did steadily increase up until the maximum TS size in both populations. Selective phenotyping was performed by setting up core sets using two approaches, average entry-to-nearest-entry distance (EN) and average accession-to-nearest-entry distance (AN). In cross-validation, in the MDL population, the highest value of prediction ability was observed on 75% population size, established using the AN approach. This method was in general more successful compared to EN in all cases, except for 20%. Starting from the size of 50%, and further decrements, average values of random sample selection of the same size had higher values of prediction ability. However, minimal values of prediction ability of random sample selection in all cases were lower compared to the AN selection approach. Similarly, in the MDL population in external validation, in general, AN showed higher values of prediction ability in all cases, except when population size was set at 75%, whereby the EN approach was more successful. The AN selection procedure in all cases had higher values of prediction ability compared to minimal values of prediction ability of multiple cycles of random sample selection. This approach in the MDL population and external validation had the highest values for a population size of 20% that contained 45 genotypes.

In the GPL population, in cross-validation, AN also had higher values of prediction ability compared to EN, except for 10% and 5% population size. Furthermore, average values of prediction ability for random sample selection were higher in comparison to the correlation rate for populations selected using the AN approach. However, minimal values of prediction ability for random sample selection were always lower compared to the AN selection methodology, implying that the best strategy for selective phenotyping would be to use the AN approach. As the training set size increased from 5% (55) to 10% (111), predictive ability increased by 0.058. The highest increase was observed when the size was raised from 10 to 20% (0.088). There were no differences in moving from a training set size of 20 to 50%. The highest values for this approach in GPL were observed for 75% population size.

In the external validation in the GPL population, the pattern of prediction ability for different methodologies was similar to that in cross-validation. However, lower correlation rates could be observed in the external validation. Prediction ability in external validation had the highest values for the smallest population of 5% (0.534), and the highest increase of 0.069 was observed when population size was increased from 10 to 20% (Figure 4d). Further increment of the population size did not increase prediction ability in the external validation.

Figure 5 graphically represents the influence of the simultaneous reduction of a number of markers and population size on prediction ability (r) in the MDL population. Based on the previous results, the RE-MoB strategy was performed for marker reduction, while the AN approach was chosen for the selection of genotypes to be included in the TS. In cross-validation (Figure 5a), the highest values of prediction ability were observed for population size greater than 50%, where population size had a larger influence compared to the number of markers. Within a population of the same size, the number of markers had little effect on prediction ability (*r_cv_*). However, it was noticed that marker subsets with 48 and 163 SNPs showed the lowest values of *r_cv_* in all combinations. For a population size of 5% and marker subsets from 546 SNPs and higher, an increase in prediction ability was detected, probably as the consequence of a low number of genotypes included in the training set and consequent overfitting. In external validation (Figure 5b), a decrease in prediction ability (*r_ev_*) was generally observed when the population size was set to 5%. The same was noticed only for the initial set of markers and the decrement in population size using the AN strategy (Figure 4c). However, this pattern was retained further on for all marker subsets. Besides that, the lowest values of *r_ev_* were detected in all populations when the marker number was 48 SNPs. Furthermore, within a population of the same size, the highest values were discovered using subsets that included 1821, 546, and 163 SNPs, depending on the specific population.

## 3. Discussion

This study compares different strategies to optimize resource allocation of genomic prediction in the soybean breeding process. Selective genotyping and phenotyping were used to optimize genomic prediction models trained for three populations with different diversity and population structures. Prediction models were developed for recombinant inbred lines (RIL) representing biparental population, multifamily diverse lines (MDL) representing specific breeding program, and germplasm collection lines (GPL). Prediction ability using different selection genotyping and phenotyping strategies was compared for the RR-BLUP method. Many studies across a wide range of traits and crops demonstrated that RR-BLUP has many advantages compared to the other statistical methods, such as relative simplicity and robustness, and the best trade-off between prediction results and efficiency in terms of computational time and power [13,28,33]. Prediction ability for yield in soybean using this method and all markers in cross-validation was 0.29 in RIL, 0.59 in MDL, and 0.72 in the GPL population, while in the external validation, the values were 0.36 in RIL, 0.60 in MDL and 0.41 in the GPL population. In other studies, prediction accuracy was higher in more closely related individuals [29] such as RIL; however, this was not the case in our research. The low value of prediction ability for yield (0.01–0.25) with an average value of 0.13 was also observed in prediction models within four soybean biparental populations [11]. In a soybean breeding program consisting of 301 experimental lines, a similar value of prediction accuracy for yield (0.64) was observed as in our MDL population [13]. Moreover, other authors observed prediction accuracies up to 0.79 for the USDA Soybean Germplasm Collection, similar to the results for the GPL population [34].

Prediction accuracy could be further improved by a reduction of the number of markers, eliminating markers with small effects or phenotypically neutral markers, as complex traits are associated with many variants that have small effects and low repeatability. Additionally, decreasing the number of SNPs can resolve the p ≫ n issue, which arises when there are significantly more features (p) than individuals (n), leading to overfitted models with poor performance [35]. This procedure also allows for decreasing computational time and resources in genomic prediction due to the reduced dimension of the data and multicollinearity between SNPs [36]. Even if a similar prediction is achieved using fewer SNPs, this can still result in the creation of novel low-density SNP chips tailored to particular traits, lowering the cost of genotyping and increasing the feasibility of genomic prediction in breeding [29]. However, the potential exclusion of SNPs that have small but cumulative effects on the trait or interact with other genetic loci can result in reduced prediction, particularly for complex traits influenced by many genetic variants with small effects [37]. Moreover, if the selection of SNPs is not conducted appropriately, there is a risk of overfitting the prediction model to the training data which can lead to poor performance when applied to independent populations [38]. Furthermore, if SNP preselection relies on prior knowledge of candidate genes, genomic regions, or marker–trait associations, in cases where such information is limited, this procedure may be less effective [39]. Several studies reported that using markers subsets could improve prediction accuracy in genomic selection in crops compared to using all SNPs [28,29,40]. The marker density should be sufficient to establish a linkage with the QTL that contributes to the variation of quantitative traits of interest. Furthermore, the selected marker set should reflect well the characteristics of all markers associated with the trait of interest and its efficiency depends on the LD between the selected SNPs and the true causative loci affecting a phenotype [41]. In some circumstances, an increase in marker density can even cause a decrease in prediction accuracy, which may be related to the increase in marker collinearity [42]. Therefore, SNP pruning was applied firstly in our study to avoid the influence of SNP clusters as markers in close positions are expected to be highly correlated.

Different strategies were used in our study to establish the smaller subsets of markers: approaches based on marker effects and model performances, without and with re-estimation, and a GWAS-based strategy. The effect of marker selection on the prediction accuracy was evaluated before in a set of soybean varieties, analyzing three strategies: random sampling, haplotype block sampling, and equidistant marker sampling [14]. These authors observed the marginal impact of marker selection on prediction accuracy for plant height. However, prediction accuracy for yield based on markers selected with a haplotype block analyses-based approach increased compared with random or equidistant marker sampling. Haplotype-based marker sampling was also performed in soybean RIL populations to compare the effects of different marker densities on the prediction of yield [11]. However, minor differences were observed among different marker densities in terms of prediction ability, probably as the consequence of a very high level of LD in soybean [11]. In the present study, the RE-MaB and NRE-MB strategies significantly increased *r_cv_* values but performed poorly in MDL and GPL populations in the external validation, indicating potential overfitting in marker selection. Conversely, the subsets obtained by applying the RE-MoB strategy can be considered the most suitable procedure for marker selection for all populations as this approach mostly led to the highest values of prediction ability that were always higher compared to minimal NRE-R values of prediction ability. The RE-MoB method identified 48 SNPs in the RIL population that hold *r_cv_* values acquired using all markers. In the MDL and GPL populations, RE-MoB chose 1821 markers that reached the plateau of prediction ability. However, in general, utilizing 546 and more SNPs did not significantly increase the *r_cv_* for yield in the MDL and GPL populations. In the external validation, the MDL population reached the highest prediction ability by selecting only 163 SNPs. A study on rice and maize datasets compared two strategies of marker preselection, using marker effects with and without re-estimation, and concluded that marker selection can improve selection accuracy recommending re-estimated effect strategies [29]. Similarly, it was confirmed that the strategy of re-estimation of marker effects improved prediction for rust disease resistance and wood density traits in Loblolly pine [26]. In our study, in most cases, the obtained results for prediction ability using different marker selection approaches were population-specific. It was observed that in the RIL population, all strategies of marker selection always led to the increment of accuracy, except the random selection of markers, while in the MDL population in external validation, marker selection led to a reduction of accuracy except the RE-MoB. In other studies, in soybean bi-parental populations were detected minimal differences in accuracy between genotyping densities of 4077 vs. 1020 SNPs [43]. Moreover, increasing the number of markers above 1000 revealed no significant improvement in prediction accuracy in an interspecific soybean NAM panel [44]. In the present study, in the GPL population, different strategies mostly led to the reduction of prediction accuracy, probably as higher marker density captured more LD in the training set for multi-generation selection. On the other hand, evaluation of a population of 235 soybean cultivars reported an increase in prediction accuracy of 4% for yield when using a haplotype block-based selection strategy for markers [14], maybe due to the lower number of genotypes present in the population. However, in a rice diversity panel including 270 elite breeding, with less related genotypes, the loss in accuracy was higher after selecting marker subsets, ranging from 11 to 38% [29]. The differences might be caused by distinct genetic characteristics between rice and soybean, influenced by their domestication history, breeding practices, and population structure. Rice typically displays lower LD compared to soybean due to its complex domestication process involving multiple independent events, leading to a diverse genetic pool. Moreover, traditional rice breeding practices have contributed to increased recombination rates and decreased LD levels [45].

An increased prediction accuracy using SNPs associated with quantitative trait loci detected by genome-wide association study (GWAS) was observed in both animals and plants [46,47]. In our study, the GWAS-based strategy developed optimal marker sets by selecting 306 and 1112 markers for the MDL and GDL populations, respectively. In the MDL population, the prediction ability of the selected marker set was lower (0.4) compared to that of the smallest set obtained with the RE-MoB approach. In the GPL population, a selected set of SNPs showed a prediction ability of 0.33, which was comparable to the observed value in models using RE-MoB sets of markers and lower compared to a high-density marker set. The reason for diminished prediction success is likely because the significant SNPs only contribute a small percentage of the total genetic variation for a phenotype [24]. Therefore, more robust methods for the selection of SNP subsets should be applied. The application of a similar method based on GWAS marker selection and random forest (RF) algorithm for genomic prediction on the soybean dataset generated from the SoyNAM population containing 5014 recombinant inbred lines (RILs) derived from 40 biparental populations [43] produced slightly higher prediction accuracies for height, time to maturity, and yield using about 10, 23, and 17% of the top SNPs, respectively [47].

Due to the continuous decrease of genotyping costs with the advance of next-generation sequencing, phenotyping is currently the most important bottleneck and needs to be optimized within plant breeding programs [19]. Good TS needs to have a population size that is both sufficient and representative of the evaluated germplasm to capture the population structure and allow for an accurate calculation of allelic effects [48]. In this study, prediction ability in cross-validation and external validation plateaued around a TS size of 113 and 553 genotypes for MDL and GPL populations, respectively. Similarly, in the model developed for the soybean breeding program, it was reported that prediction accuracy increased as TS size increased and that a plateau in yield prediction was reached at around 100 breeding lines [13]. Probably, as TS size increased, rare allele frequencies increased, which helped to improve estimations of marker effects [13]. Moreover, it is considered that increasing the TS size is advantageous to enhance genetic gains when the TS is composed of genotypes being phenotyped and selected over generations [42]. In addition, multiple studies have shown a wide variation in the optimum population size for soybean prediction models [11,43,44]. Namely, testing the SoyNAM panel with more than 5000 individuals in multiple environments showed a plateau for accuracy at 2000 individuals across several traits, grain yield, days to maturity, plant height, pod number, node number, and pods per node [43]. Furthermore, in an interspecific soybean NAM panel, the predictive ability for yield increased by 67% with a training set size of 50 to 350 individuals, with no significant gains above 250 individuals in the TS [44]. The predictive ability for yield increased significantly as the TS size increased from 50 to 300 genotypes, in the set including 483 RILs from 26 pedigrees [11]. Overall, the specific TS size for genomic selection might be population-dependent due to varying levels of genetic structure and LD [11].

The most common selective phenotyping is based on random sampling, mainly because of its simplicity and generality. However, this approach could be inappropriate as it might not reflect true relationships in the original population, especially in the presence of a strong population structure [31]. As a result, certain genotypic diversity components will be under-represented in the training set, while others will be over-represented, adversely affecting predictive ability. The application of an adequate selective phenotyping procedure should reduce the cost of phenotyping while still capturing genetic variation relevant to the trait of interest and maintaining the high accuracy of prediction models. This also optimizes the allocation of limited resources, thereby enhancing the efficiency of GS programs. However, reduced representation of genetic variation in cases when rare alleles or specific genetic variants are not present in the TS could potentially reduce the accuracy of genomic prediction, particularly for traits influenced by rare variants [49]. Moreover, introduced sampling bias when TS does not adequately represent the genetic diversity present in the entire population, leads to biased estimates of GEBVs [50]. Different studies examined a wide range of TS optimization methods including the mean of the coefficient of determination (CDmean), the mean of the prediction error variance (PEVmean), stratified sampling, partitioning around medoids (PAM), Rscore, generalized average genomic relationship (gAvg_GRM), and random sampling [51,52,53]. In general, it was concluded that a training set size of around 50–55% of all available genotypes usually generates accuracies in the range of 95–100% of the maximum for targeted optimization that use the information from the test set to build TS, while for untargeted optimization that does not use genomic information from a test set to determine the training set, a TS size of 65–85% is required for similar results [52]. The general conclusion is that there are no universal criteria for genotype selection, yet it mainly depends on LD between markers, the relationship between training and test set, the genetic architecture of the trait, heritability, and population structure [51,52]. Our research performed selective phenotyping for the MDL population and GPL population by setting up core sets using two approaches, average entry-to-nearest-entry distance (EN) and average accession-to-nearest-entry distance (AN). In cross-validation, the highest prediction ability for the MDL population was observed at 75% population size (170 genotypes) using the AN approach. GPL population, in general, had lower correlation rates in external validation compared to cross-validation for different methodologies. In both populations, the AN strategy was more successful compared to EN. Furthermore, the average prediction ability for random selection was higher compared to the AN approach, while minimal values of prediction ability of random selection were always lower compared with AN, implying that the best strategy for selective phenotyping would be the AN approach. While the AN approach tends to give cores that maximally represent all individual accessions from the original dataset, focusing on cluster centers, the EN approach selects core accessions to represent the whole range of values from the original dataset, including the extremes, which might be the reason for better prediction power of AN strategy. Similarly, the generalized average genomic relationship (Avg_GRM_self) algorithm was suggested as an optimization technique in the untargeted scenario because it computes quickly and scales well for datasets with high dimensionality, and it consistently demonstrated the best performance across datasets (maize, rice, sorghum, soybean, spruce, and switchgrass) and traits regardless of population structure [52]. The goal of this method, which is based on a relationship matrix, was to minimize the relationship inside the TS.

## 4. Materials and Methods

### 4.1. Material

In this study, three different soybean datasets were considered for the evaluation of genomic prediction models for grain yield.

1. Recombinant inbred lines (RILs) population selected from nested association mapping (NAM) panel was selected to simulate genomic prediction models developed within the RIL population. This study investigated the NAM14 population containing 137 F5-derived RILs from the cross between high-yielding parents IA3023 and Magellan. Phenotypic and genotypic data were extracted from the R package SoyNAM available on CRAN (https://cran.r-project.org/web/packages/SoyNAM/index.html, accessed on 20 September 2020). The lines were grown across nine environments in four USA states (Iowa, Illinois, Indiana, and Nebraska) from 2011 to 2013. Phenotypic data for yield were expressed as the best linear unbiased prediction (BLUP), calculated using the R package lme4 (version 1.1-27) [54]. The external validation population consisted of 20 randomly selected genotypes that were evaluated in three environments (Illinois_2013, Indiana_2012, and Nebraska_2011), while the TS for external validation consisted of the remaining 117 genotypes evaluated across six environments (Iowa_2012, Iowa_2013, Illinois_2011, Illinois_2012, Indiana_2013, and Nebraska_2012). RILs from NAM14 were genotyped using the Illumina Infinium SoyNAM6K BeadChip [55]. Missing marker data were imputed using an algorithm implemented in the software Beagle v4.1 [56]. A total of 3088 SNPs were successfully detected within the NAM14 population.

2. Multifamily diverse lines (MDL) consisted of soybean lines and varieties obtained from the existing breeding programs of the Institute of Field and Vegetable Crops (Serbia), previously described [15]. The population comprised a set of 227 soybean genotypes (F_4:6_ lines). Genotypes from the MDL training set were evaluated for seed yield on experimental fields of the Institute of Field and Vegetable Crops (Rimski šančevi, Serbia) in 2014–2016. BLUP of soybean genotypes was estimated by fitting the model assuming random genotypic effect and environmental effects and was performed using the lme4 package in the software R, version 4.1.0 [54]. Obtained BLUP values were used for the development of the genomic prediction models. The validation population used for external evaluation of the prediction model originated from the same breeding program and was composed of 21 elite soybean varieties. External validation of models was performed based on yield data collected in trials over 7 years (2010–2016), in 10 to 14 locations per year in the province of Vojvodina (Serbia). The MDL population was genetically characterized using a genotyping-by-sequencing approach with the ApeKI restriction enzyme [57]. From the initial marker set, SNPs with PMV ≤ 80% and MAF > 0.05 were removed, leaving 34,889 SNP markers for further analysis. Missing SNPs were imputed based on an algorithm implemented in the software Beagle v4.1 [56].

3. Germplasm collection lines (GPL) were included to represent the most comprehensive model. The GPL set included 1107 PIs from the USDA soybean germplasm collection and consisted of 163, 41, 17, 36, 129, 512, 123, 31, and 55 accessions of maturity groups (MG) 0-VIII, respectively. The phenotypic data used in this study were obtained from the USDA Soybean Germplasm Collection evaluation [34]. The external validation population included 301 accessions. USDA soybean germplasm collection had been genotyped using the Illumina Infinium SoySNP50K BeadChip [58] and the SNP data are publicly available at Soybase (http://www.soybase.org/dlpages/index.php, accessed on 10 October 2020). SNPs with PMV ≤ 80% and MAF > 0.01 were removed from the dataset, leaving 38,184 SNPs for selected genotypes. Missing data were imputed using Beagle v4.1 [56].

### 4.2. Methods

#### 4.2.1. Genomic Prediction

Genomic prediction modeling was performed using ridge-regression best linear unbiased prediction model (RR-BLUP). RR-BLUP is a mixed model that assumes additive effects of markers with equal genomic variances and where marker and residual effects are randomly distributed. This approach follows a linear form:y = µ + Xβ + e,(1)
where y is a t × 1 vector of phenotypic data (t is the size of the training set), µ is the overall mean, X is the marker–genotype matrix (t × m; m is the number of markers), β is a m × 1 vector of marker effects, and e is the residual error. In external validation, estimated marker effects were used to predict the GEBVs of the validation population, calculated as µ + X_1_ β, where X_1_ is a marker–genotype matrix of the validation population (v × m; v is the size of the validation population). RR-BLUP was conducted using the rrBLUP R package (version 4.6.1) [59].

Genomic prediction models were evaluated based on prediction ability, calculated as Pearson product–moment correlation coefficient (r) between the observed phenotypic data and GEBVs estimated by the prediction model. The efficiency of genomic prediction models was validated through the following:Cross-validation with 20 repetitions of a 10-fold scheme, where prediction ability (*r_cv_*) was evaluated with a validation set representing 10% of genotypes from the training set;External validation, where prediction ability (*r_ev_*) was evaluated based on genotypes that were not included in the training set.

#### 4.2.2. Selective Genotyping Strategies

##### SNP Pruning

Filtered SNP datasets were pruned using the SNPRelate R package (version 1.34.0) [60]. The aim was to obtain a set of SNPs that are in approximate linkage equilibrium with each other, to avoid the influence of SNP clusters. For linkage disequilibrium (LD) pruning, an r^2^ threshold of 0.95 was set, so that SNPs in pair-wise comparison within the window that had a squared correlation above the threshold were removed.

##### Approaches without and with Re-Estimation of Marker Effects

Following pruning, two approaches were performed to reduce the number of markers: (1) without re-estimation of marker effects (NRE) and (2) with re-estimation of marker effects (RE). For the (NRE) approach, marker effects of the pruned set were estimated following the RR-BLUP method (NRE-MB = no re-estimation marker-based). Markers were ranked in decreasing order based on the absolute values of marker effect to select 30% of SNPs with the highest absolute value. This step was repeated until a subset with 48 SNPs was identified (6072, 1821, 546, 163, 48 SNPs). In addition, the same-sized sets were obtained by randomly selecting 6072, 1821, 546, 163, and 48 markers (NRE-R = no re-estimation random markers). In the second (RE) approach, the effects of pruned markers were estimated and ranked in decreasing order based on absolute values. Firstly, a set of 30% of markers (6072) with the highest absolute value was selected (RE-MaB = re-estimation marker-based) (Appendix A). The model was recreated based on this set of markers and the RR-BLUP method and the marker effects were recalculated. Markers were ranked based on absolute values and again top 30% of markers (1821) were selected. This step was repeated until a subset with 48 SNPs was identified.

The next performed strategy with re-estimation was model-based (RE-MoB = re-estimation model-based) (Appendix A). This approach was based on a random selection of the specific number of markers that were used to make models in 1000 repetitions. Based on the prediction ability of 1000 models, the best model was selected and corresponding markers became the new entry marker set. This procedure was repeated for a specific number of makers previously defined. For marker sets obtained with all strategies, cross-validation and external validation were performed and GEBV and r^2^ values were calculated.

##### Genome-Wide Association Study (GWAS) Based Strategy

Marker selection was also performed based on GWAS results in the R package GMStool version 1 [24]. This tool selects the optimal marker set using statistical and machine/deep-learning models for genomic prediction and suggests the best prediction model with the optimal marker set. GMStool starts construction of the marker set with SNP markers with the lowest *p*-value based on GWAS results and continues by adding markers that increase prediction accuracy. In the phase of marker selection, RR-BLUP and bootstrap trees (BTS) methods are used as learning models. Afterwards, GMStool performs the final modeling phase using different methods (RR-BLUP, random forest (RF), deep neural network (DNN), convolution neural network (CNN)) and the best prediction model and optimal marker set are selected depending on the correlation rate between the observed and predicted phenotypes of the validation set. GWASs were performed only for the training sets (MDL and GPL populations) using a multi-locus, modified mixed linear model (MLMM) incorporated into FarmCPU [32].

#### 4.2.3. Selective Phenotyping Strategies

The reduction of the number of genotypes was performed by selecting a core set from the initial population in the R package Core Hunter 3 [61]. Genetic distance and kinship matrix between pairs of soybean genotypes were computed using the identity-by-state (IBS) method implemented in TASSEL5 [62]. The obtained distances were utilized for creating core sets in package Core Hunter 3, sampling 75, 50, 20, 10, and 5% of genotypes from the initial set. For the identification of core sets, two approaches were used:

1. Average entry-to-nearest-entry distance (EN) where the mean distance between each selected accession and the closest other selected accession was evaluated. Maximizing this measure yields high diversity in the core, where each selected accession is sufficiently different from the most similar other selected core accessions.

2. Average accession-to-nearest-entry distance (AN) where the mean distance between each accession in the whole collection and the closest selected accession was evaluated. Minimizing this measure yields cores that maximally represent all individual accessions from the original dataset.

## 5. Conclusions

Genotyping and phenotyping selection strategies have good potential and can be used to reduce the number of markers and genotypes, taking into account the specificity and characteristics of the population of interest. In general, our study confirmed that the number of markers had less impact on predictive ability compared to TS size, and within the population of the same size, marker density had a low impact on prediction ability. Thus, the application of a lower number of markers to improve or at least retain the accuracy of genomic selection obtained using a high-density marker set is a realistic possibility that could significantly reduce genotyping costs. For marker selection across all analyzed populations, the subsets produced by the model-based approach can be regarded as the most appropriate. Furthermore, the markers-based selective phenotyping consistently had greater values compared to the minimal values of random selection.

## Figures and Tables

**Figure 1 plants-13-00975-f001:**
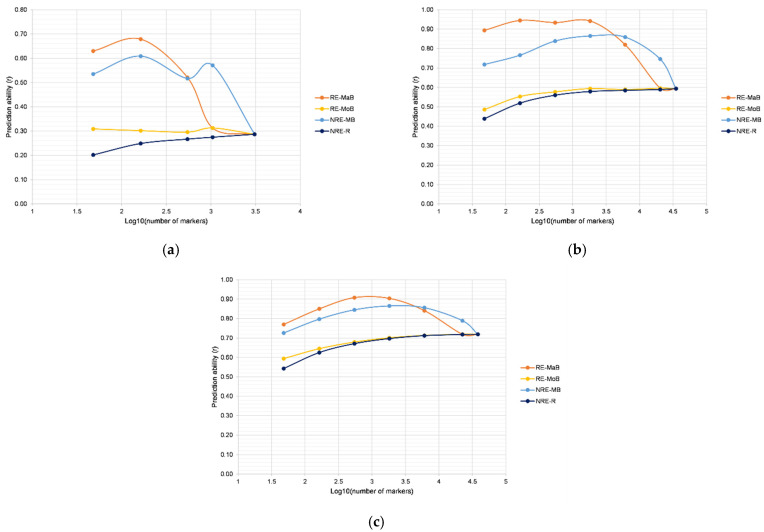
Cross-validation in: (**a**) RIL, (**b**) MDL, and (**c**) GPL populations using different strategies for marker selection.

**Figure 2 plants-13-00975-f002:**
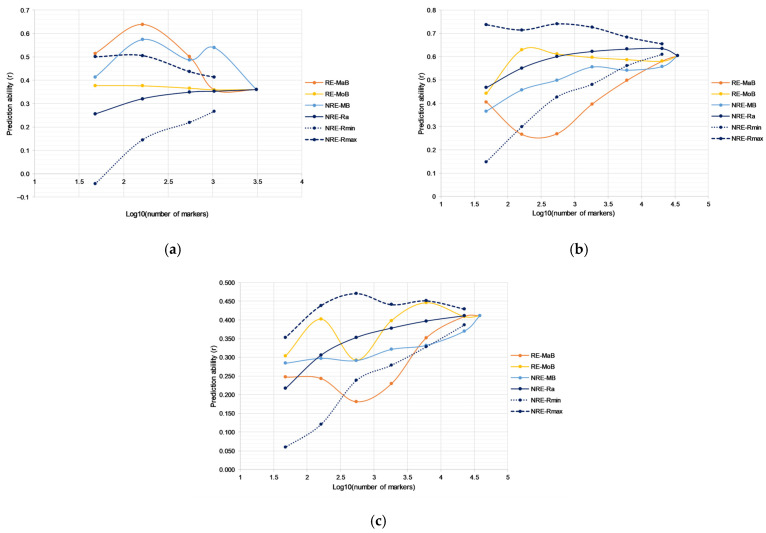
External validation in: (**a**) RIL, (**b**) MDL, and (**c**) GPL populations using different strategies for marker selection.

**Figure 3 plants-13-00975-f003:**
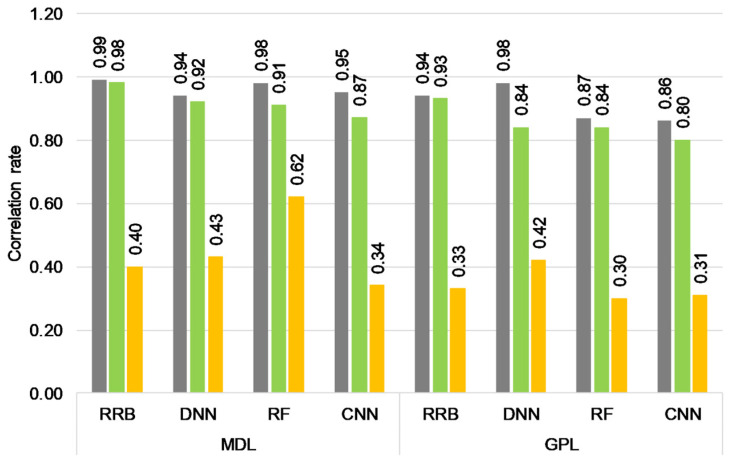
Correlation of observed and predicted phenotypes of training (grey), validation (green), and test (orange) sets in the MDL and GPL populations using different prediction models.

**Figure 4 plants-13-00975-f004:**
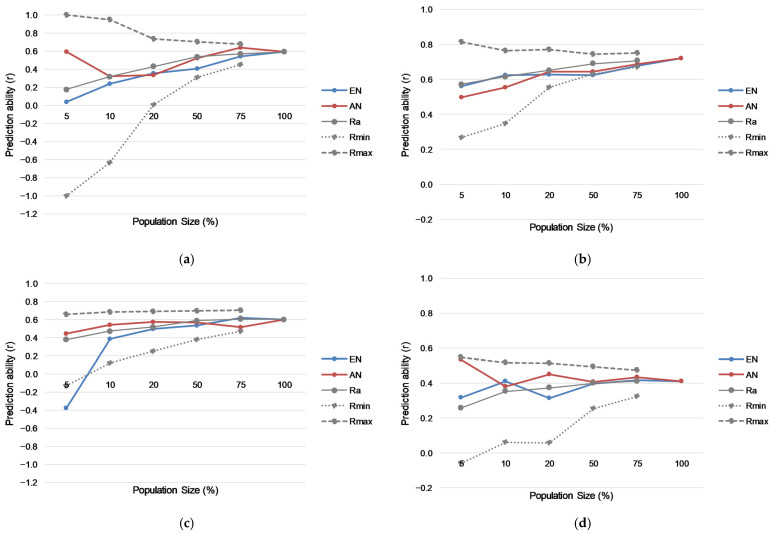
Prediction ability in different sized training sets in cross-validation for: (**a**) MDL population and (**b**) GPL population and in external validation for: (**c**) MDL population and (**d**) GPL population (Ra—average values of random sample selection, Rmin—minimal values of random sample selection, Rmax—maximal values of random sample selection).

**Figure 5 plants-13-00975-f005:**
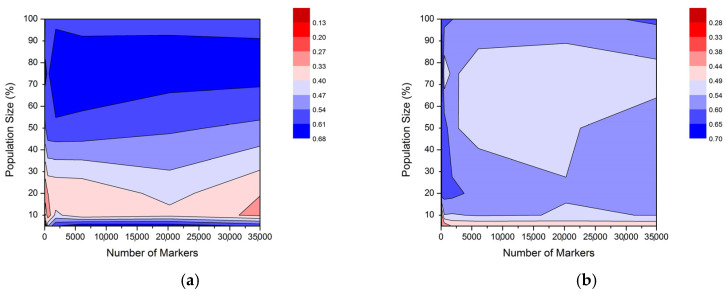
Influence of simultaneous reduction of marker number and population size on prediction ability (r) in MDL population in (**a**) cross-validation and in (**b**) external validation.

**Table 1 plants-13-00975-t001:** Population size and number of markers for evaluated populations.

Population	Training Set Size (CV)	Training/Validation Set Size (EV)	Initial SNPs	Pruned SNPs
RIL	137	117/20	3088	1044
MDL	227	227/21	34,889	20,242
GPL	1107	1107/301	38,184	22,470

**Table 2 plants-13-00975-t002:** Sample size, selected marker sets, and correlation rates for CVs for the MDL and GPL populations.

Population	Sample Size			Markers		Correlation Rate (Mean ± Std.)	
	Training	Validation	Test	All	Selected	Training	Validation
MDL	151	76	21	34,889	306	0.948 ± 0.015	0.941 ± 0.058
GPL	886	221	301	38,184	1112	0.917 ± 0.043	0.886 ± 0.070

## Data Availability

Publicly available datasets were analyzed in this study. These data can be found here: https://cran.r-project.org/web/packages/SoyNAM/index.html, accessed on 20 September 2020, http://www.soybase.org/dlpages/index.php, accessed on 10 October 2020.

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
