# Peer review of "Selective Genotyping and Phenotyping for Optimization of Genomic Prediction Models for Populations with Different Diversity"

_plants, 2024, doi:10.3390/plants13070975_

Round 1

Reviewer 1 Report

Comments and Suggestions for Authors

This study aimed to evaluate methods of selective genotyping and phenotyping for soybean yield prediction. Different strategies for marker selection were tested, with model-based strategies showing the most promising. Selective phenotyping based on markers consistently outperformed random selection. The results of this study seem interesting and can be implemented by other scientists in different plant species. However, there are some very major and important points needed to be considered before the manuscript is accepted:

Title: you can remove one “selective” word from the title and make it more concise and right to the point.

Some of the keywords are already presented in the title. Please use different words than the title.

Line 21-22 how the selection conducted? I am not sure whether the core set was diverse as possible to be used well in training GS.

Line 25, what do you mean by “higher value”? Please provide some data in the abstract quantifying what you have mentioned there.

The introduction part needs to be significantly improved. The authors need to explain more about the recent efforts in genomic selection using soybean population. There are good references for that, such as https://link.springer.com/article/10.1007/s11032-018-0872-4,  https://link.springer.com/article/10.1007/s11032-016-0504-9, and https://www.cell.com/heliyon/fulltext/S2405-8440(22)03161-9?_returnURL=https%3A%2F%2Flinkinghub.elsevier.com%2Fretrieve%2Fpii%2FS2405844022031619%3Fshowall%3Dtrue.

The discussion part needs to be improved. You need to explain more about your research and compare the results with previous studies. Also, bring the strengths and the limitations of this research into the discussion and discuss them.

Please check the paper for some grammatical and punctuational errors.

Line 459 10-fold CV with how many repetitions? 

Author Response

We would like to thank the Reviewer for useful comments and suggestions. We tried to incorporate all of them into the manuscript. Here are the responses to the reviewers' comments:

  1. Title: you can remove one “selective” word from the title and make it more concise and right to the point.

The reviewer's comment has been acknowledged and the requested change was made. The title (L2-4) now reads as:

Selective genotyping and phenotyping for optimization of genomic prediction models for populations with different diversity

  1. Some of the keywords are already presented in the title. Please use different words than the title.

The proposed changes have been made and a new set of keywords is (L35-36): soybean yield; genomic selection; single nucleotide polymorphism; training set; soybean breeding

  1. Line 21-22 how the selection conducted? I am not sure whether the core set was diverse as possible to be used well in training GS.

More information is provided in the Material and Methods section. Genetic distance and kinship matrix between pairs of soybean genotypes from the initial population were utilized for selecting core sets of genotypes in package Core Hunter 3, using two approaches. The first approach, average entry-to-nearest-entry distance (EN), selects an accession for a core set to be sufficiently different from the most similar other selected core accessions. The second approach, average accession-to-nearest-entry distance (AN), yields cores that maximally represent all individual accessions from the original dataset.

We have expanded the sentence in the abstract to clarify this and it now reads as (L25-27): “Reduction of the number of genotypes is performed by selecting a core set from the initial population based on marker data, yet maintaining the original population's genetic diversity.”

  1. Line 25, what do you mean by “higher value”? Please provide some data in the abstract quantifying what you have mentioned there.

“Higher value” refers to the higher value of prediction ability. We have made corrections in the abstract and the sentence now reads as (L29-32): ”The selective phenotyping based on makers in all cases had higher values of prediction ability compared to minimal values of prediction ability of multiple cycles of random selection, with the highest values of prediction obtained using AN approach and 75% population size.“

We have also made similar changes throughout the whole manuscript to improve clarity.

  1. The introduction part needs to be significantly improved.

The authors need to explain more about the recent efforts in genomic selection using soybean population. There are good references for that, such as https://link.springer.com/article/10.1007/s11032-018-0872-4  ,  https://link.springer.com/article/10.1007/s11032-016-0504-9 , and https://www.cell.com/heliyon/fulltext/S2405-8440(22)03161-9?_returnURL=https%3A%2F%2Flinkinghub.elsevier.com%2Fretrieve%2Fpii%2FS2405844022031619%3Fshowall%3Dtrue.

The reviewer's comment has been acknowledged and the requested change was made. In the introduction section, we have added more relevant information related to soybeans (L39-47) and recent efforts in genomic selection using soybeans (L51-56).

  1. The discussion part needs to be improved. You need to explain more about your research and compare the results with previous studies. Also, bring the strengths and the limitations of this research into the discussion and discuss them.

We appreciate your valuable comment regarding the discussion section of the manuscript. We have made changes throughout the discussion to address these points accordingly to enhance the quality of the manuscript. We provided a more comprehensive explanation of our research findings, comparing them with relevant previous studies. Additionally, we have included the strengths and limitations of selective genotyping and phenotyping approaches to provide a well-rounded evaluation of the study.

  1. Please check the paper for some grammatical and punctuational errors.

We appreciate the reviewer's remarks. Changes have been made throughout the text to correct grammatical and punctuational errors.

  1. Line 459 10-fold CV with how many repetitions? 

A 10-fold CV was performed in 20 repetitions. However, for random marker and phenotype selection, the repetitions were performed in 100 and 1000 repetitions, but it is clearly stated in the text in the Material and Methods section.

We have added this in the manuscript and the sentence now reads (L595-597) as: “…cross-validation with 20 repetitions  of a 10-fold scheme, where prediction ability (rcv) was evaluated with a validation set representing 10% of genotypes from the training set…”

Reviewer 2 Report

Comments and Suggestions for Authors

The study uses soybean phenotypic and SNPs genotypic data collected from biparental population, multifamily diverse lines representing specific breeding program, and germplasm collection. Different strategies to optimize resources allocation in a soybean breeding process were investigated, including the effects of selective genotyping and of selective phenotyping on the selection efficiency estimated through the prediction ability, cross-validation and external validation process. The results confirmed previous findings on other crops and provide further insights into the prediction ability using different marker selection approaches and the influence of the population.

The study, which is analytical in nature, is founded on established methodologies and has resulted in new incremental information to warrant publication. The quality of the presentation is also acceptable.

Author Response

Thank you for your comments and for taking the time to review our work. Your feedback is greatly appreciated.

Reviewer 3 Report

Comments and Suggestions for Authors

This is a fine manuscript.  I especially appreciated the emphasis on predictive accuracy, the use of cross-validation, and the reduction of overfitting and of false positives and false negatives.  That makes for solid results.

Author Response

(The authors gave the same response as above.)

Round 2

Reviewer 1 Report

Comments and Suggestions for Authors

All of my comments were addressed.